# Rheological and Durability Properties of Self-Compacting Concrete Produced Using Marble Dust and Blast Furnace Slag

**DOI:** 10.3390/ma15051795

**Published:** 2022-02-27

**Authors:** Cenk Karakurt, Mahmut Dumangöz

**Affiliations:** 1Department of Civil Engineering, Bilecik Seyh Edebali University, Bilecik 11230, Turkey; 2Department of Construction Technology, Aksaray University, Aksaray 68100, Turkey; mahmutdumangoz@aksaray.edu.tr

**Keywords:** durability, granulated blast furnace slag, marble dust, rheology, self-compacting concrete

## Abstract

Self-compacting concrete (SCC) is a special, highly fluid type of concrete that is produced using chemical additives. It is easier to pour and reduces defects arising from workability. Waste marble dust is generated during the production of marble using different methods, or during the cutting of marble in processing plants; however, the uncontrolled disposal of waste marble dust in nature is associated with some environmental problems. Cement and concrete technology is a field with potential for the utilization of these large amounts of waste. The present study explores the use of marble dust (MD) (an industrial waste generated in abundance around the province of Bilecik) and granulated blast furnace slag (GBFS) (another industrial waste product) in the production of SCC. In this study, MD and GBFS are used as fine materials in SCC mixtures, and the rheological and workability properties and other hardened concrete properties of the produced SCC specimens are tested. Additional tests are conducted to identify the durability of the specimens to sulfate attack, as well as their freeze–thaw and abrasion resistance, followed by microstructure tests to identify the effects of MD and GBFS on bond structure. The late-age performances of MD and GBFS were then examined based on the results of the durability tests. The presented results revealed improvements in the fresh and hardened properties of SCC produced using MD and GBFS.

## 1. Introduction

Concrete is one of the most important and accessible construction materials in the market. The fresh and hardened state of this cementitious composite plays an important role in the workability, strength, durability and labor costs of the construction. Self-compacting concrete (SCC) is a special type of concrete with a high rate of consistency that can be spread and consolidated under its own weight within the framework; that is leveled through the expulsion of trapped air and compacting, even in the most heavily reinforced areas and narrowest sections, without the need for vibration; that does not cause such problems as segregation and bleeding; and that maintains its cohesion [1,2,3].

SCC was first developed in Japan in the late 1980s as a result of efforts to develop more durable concrete [4]. SCC is highly fluid and can be spread over the desired section under its own weight, achieving fullness in the section without the need for internal or external vibration [5]. In addition to the above-mentioned characteristics, SCC is required to resist either blockage or clogging between obstacles when flowing, as well as segregation [6]. These characteristics are associated with the fluidity, permeability and segregation resistance of SCC [7,8]. The use of high-performance plasticizers in particular can increase workability without impairing the viscosity or cohesion of the fresh concrete, which can lead to segregation [9]. The two main features of SCC are its high fluidity and its high resistance to segregation, which are achieved through the use of large amounts of powder materials and plasticizers. The aim of using plasticizers is to obtain the desired rheological behavior of the fresh concrete according to the technology used and the conditions for the implementation process of concreting [10]. Fly ash, stone dust, GBFS and silica fume can all be used as viscosity-enhancing powder materials in SCC [11].

The rheological or flow properties of concrete are important due to their role in the placement, strength and durability of the product. Fresh concrete exhibits fluid behavior and is often assumed to have the properties of a Bingham fluid. In such a fluid profile, at least two parameters, including threshold (critical) shear stress and viscosity, are used to describe the flow behavior [12]. It is noted that in some cases, however, shear thickening can occur due to high shear rates or low water content, as shown in Figure 1 [13,14]. Previous studies have shown that the mineral additives used in the mixture, such as quartz or fly ash, as well as the particle size distribution, contribute to shear thickening [15,16]. In such cases, it has been noted that shear thickening can be better expressed by Herschel–Bulkley and modified Bingham models [17].

With the industrialization of society, the management of waste generation has become one of the most serious environmental problems. In order to overcome harmful environmental pollution, appropriate waste management becomes a resource that contributes to the reduction of raw material consumption, provides significant benefits in the protection of natural assets, reduces the impact on climate change and supports sustainable development [18]. Mineral-based materials such as fly ash, metakaolin, GBFS, silica fume and stone dust can be used as powder components in SCC mixtures [19], and there have been studies demonstrating their positive effects on workability, mechanical properties, shrinkage and crack formation [20,21]. Meera et al. found that equivalent workability could be achieved through the use of fewer plasticizer additives in SCC mixtures containing granite and marble powder than in those containing fly ash [22]. The fine MD generated during the cutting of marble can be used in the production of concrete to increase the filler effect of fine aggregate, and marble waste is used as an alternative to limestone in the cement sector. There are over 200 marble cutting plants in Bilecik, making the utilization of marble waste essential. Marble wastes can be generated from the beginning of the mining process and machining processes such as cutting, polishing and finishing of this natural stone [23]. The use of MD in concrete reduces the permeability of the concrete and improves its mechanical properties [24]. In their study, Gameiro et al. reported improvement in the water absorption and capillary action of concrete containing 20% marble aggregate, thereby achieving higher durability [25]. Especially, depending on the fine particle size distribution of the MD, it exhibits a favorable inner filler effect for the improved workability performance. Tennich et al. concluded that the marble wastes are beneficial and have a positive effect on the fresh state of the concrete by reducing plastic shrinkage [26]. In addition, the higher calcium content of MD helps to strengthen the hardened properties of the SCC [27].

GBFS is an important pozzolanic material that is generated as a by-product in the iron and steel sector. It becomes glassy when cooled rapidly and can be finely ground for use as a mineral additive in concrete and cement [28,29]. Research has shown that large proportions of GBFS, when used as a mineral additive, can have positive effects on the durability of reinforced concrete structures that are exposed to sulfate attack, and reduce the chloride permeability of concrete [30,31]. The use of GBFS as a powder component in SCC mixtures offers many advantages, with studies having shown the long-term preservation of compactability and concrete consistency in SCC, as well as improvements in durability. As to the rheology of fresh concrete, it is noted that particle size optimization and particle packing are improved through the use of GBFS, thus increasing its segregation resistance [32]. Improvement was observed in workability when GBFS was used in place of cement in proportions up to 20%; however, the increased slag content resulted in decreased yield stress and plastic viscosity [33]. As to the properties of hardened concrete, late-age strength increases, while the use of GBFS at proportions of 50–75% in SCC mixtures has been shown to have a positive effect on the durability of concrete [34].

This study aims to propose a different field in the economic recovery of these industrial wastes by examining the performance of MD and GBFS as dust content in SCC mixtures. In the present study, the rheological and durability properties of SCC mixtures containing MD and GBFS were determined through tests, with fresh concrete and rheological properties being analyzed when using MD and GBFS additives in different proportions in place of fine aggregate in fresh SCC mixtures. The study explores the effect on the compressive strength and the behavior of SCC to abrasion effects, freeze–thaw and sulfate attack, as the most common phenomena to which hardened SCC is subjected during its lifecycle. The results showed that the mechanical and durability properties of the SCC can be increased by utilizing MD and GBFS up to 10% and 30%, respectively.

## 2. Research Gap

With the global acceptance of the Paris Climate Agreement, waste management and the establishment of a sustainable ecosystem have become important issues. According to this case, sustainable and greener production of construction technologies and eco-friendly evaluation of industrial by-products are current issues [35]. There have been lots of studies about the usage of industrial by-products as a replacement material against cement for the production of SCC [36,37,38,39,40]. This study is focused on the fine aggregate replacement effect on the fresh and hardened properties of SCC mixtures. Limited studies have been carried out about the replacement of fine aggregate against industrial wastes [41] as investigated here for SCC mixtures. The effect of MD and GBFS as powder waste on the filling effect in both fresh and hardened states of SCC, especially the effect of powder materials on the fresh properties of SCC, is focused by comparing the standard SCC consistency tests and rheological results. Additionally, the pozzolanic nature of GBFS in SCC mixtures has an additional benefit for the strength development of the hardened SCC specimens besides the filler effect of the powder.

## 3. Materials and Methods

### 3.1. Materials

Ordinary CEM I 42.5R Portland Cement produced by the SANÇİM Cement Plant Bilecik, Turkey was used in the study, while tap water in Bilecik was used as the mixing water in the production of the mortar and concrete specimens. In the SCC mixtures, the largest aggregate particle size is required to be below 20 mm [42]. The aggregates used in the specimens were limestone-based crushed stone aggregate in three different grain classes, namely, 0–4 and 4–12 mm, obtained from the Dağ-İş Mining Company Bilecik, Turkey. A polycarboxylic ether-based water reducer BASF GLENIUM C 303 was used as a plasticizer in the mixtures. The powder additives used in the production of the SCC samples were GBFS, obtained from the ERDEMIR Iron and Steel Plant, Ereğli, Turkey, and MD, obtained from the HARTAŞ Bilecik, Turkey marble plant in the form of cutting waste sludge, which was first dried and then sieved through a 0.125 mm sieve.

### 3.2. Method

The specific surface area and density of the raw materials used in the mixtures were determined using Brunauer, Emmet and Teller (BET) and gas pycnometer devices manufactured by Micrometrics, and the phase and chemical analyses were conducted with X-ray fluorescence (XRF) and X-ray diffraction (XRD) devices manufactured by Panalytical. To determine the optimum SCC mixing ratios, trial castings were carried out according to the methods described in European Federation for Specialist Construction Chemicals and Concrete Systems (EFNARC) [42]. Rather than a 0–4 mm aggregate, MD and GBFS were used in the mixtures at ratios of 10, 20 and 30 percent of the total amount of powder. The largest aggregate grain size in the SCC mixtures was 12 mm. The amount of plasticizer in all mixtures was 2% by weight of cement. The mixing ratios of the concrete specimens used in the tests are presented in Table 1. The workability properties of the SCC mixtures were determined with slump-flow [43], V-funnel [44], and L-box [45] test devices produced by Utest Laboratory Device Company, while the rheological parameters of the concrete were investigated with a BT-2 compact fresh concrete rheometer manufactured by Schleibinger. This rotational rheometer had two probes measuring the torque during the rotation inside the concrete that was placed without any segregation. Thus, relative viscosity and relative yield stress can be determined by the rheometer.

The mechanical properties of the hardened concrete specimens were determined through a uniaxial compressive strength test on 150 × 150 × 150 mm cube specimens conditioned in a lime-saturated curing pool under standard curing conditions for 7, 28 and 56 days [46]. A Bohme abrasion resistance test [47] was performed on 70 × 70 × 70 mm cube samples to determine the abrasion resistance of the SCC. The weight loss of the cubic specimens was determined at the end of 440 cycles of the steel plate in diameter with 750 mm. The abrasion effect was increased with abrasive corundum dust usage during the rotation of the steel plate. Additionally, a freeze–thaw test was conducted at 50 cycles on 100 × 100 × 100 mm specimens at temperatures between −20 °C and 20 °C [48], and a sulfate resistance test [49] was performed on 25 × 25 × 285 mm specimens. The mortar bar specimens were immersed in Na_2_SO_4_ solution to determine the length changes at the end of 180 days exposure time. Changes in the internal structures of the specimens were identified using a Zeiss SUPRA 40VP scanning electron microscope (SEM). The schematic representation of the performed tests is presented in Figure 2 below.

## 4. Results

### 4.1. Material Characterization

Limestone-based crushed stone aggregates of two different grain size classes were used in the SCC mixtures. The granulometric curve obtained from the sieve analyses of the aggregates, in which the largest aggregate particle size was 12 mm, is shown in Figure 3.

The chemical properties of the cement and mineral additives used in the SCC mixtures were determined by performing XRF analysis, the results of which are given in Table 2. It can be seen from the table that the cement and the GBFS had a high content of SiO_2_ and CaO, while the MD was predominantly CaO. The SiO_2_ and Al_2_O_3_ content of the GBFS suggests the existence of the pozzolanic potential of the material. The reactive silicate content of the GBFS was found to be 48.74% due to the method prescribed in TS EN 196-2 [50]. The loss on ignition (LOI) of MD is high due to the dissociation of CaCO_3_ and dolomite that partially transform themselves into CO_2_ at higher temperatures during the LOI test.

The mineralogical structure of the cement, GBFS and MD used in the mixtures were determined by an XRD analysis, the results of which are presented in Figure 4. Figure 4a illustrates the distribution of the main component phases of the CEM I 42.5 R Portland cement used in the mixtures. The alite and belite phases, which contribute to the strength of the cement through hydration, were found in high amounts in the chemical structure of the cement. The high C_3_S content (Table 2 and Figure 4a) plays a key role in increasing the early strength properties of the cement. As to the mineralogical structure of the MD (Figure 4b) used as a mineral additive in the SCC mixtures, the body is generally composed of calcite and dolomite phases, which is associated with the presence of CaCO_3_ in the chemical structure of the marble. The results of the XRD analysis of the GBFS specimen clearly indicate the presence of an amorphous structure, and no sharp peaks could be obtained (Figure 4c). The reactive SiO_2_ and CaO content of the GBFS are thought to affect the performance of hardened SCC mixtures [51]. The high reactive SiO_2_ content of GBFS is related to the proper cooling of molten slag with an amorphous structure as presented in Figure 4c.

The results of the BET-specific surface area and gas pycnometer density tests conducted on the powder materials used in the production of SCC are given in Table 3, revealing the mineral additive with the finest grains to be MD. The specific surface area of the GBFS, which is harder and more difficult to grind than other granular materials, has a lower specific surface area than cement and MD. As to the material densities, the MD was found to be lighter than GBFS.

### 4.2. Fresh Concrete Tests

#### 4.2.1. Workability Tests

The workability properties of the SCC mixtures were determined by slump flow, V-funnel and L-box tests, the results of which are given in Table 4, and indicate a change in workability with the increase in the ratios of GBFS and MD additives. In general, the SCC mixtures were observed to be within the limit values given in EFNARC. The slump test results varied between 550 mm and 720 mm. While the MD30 series was in the SF2 class, the other mixtures were in the SF1 class. The best slump value was achieved by the GBFS20 mixture at 720 mm, corresponding to a 7.5% better flowability than the Reference.

The results of the V-funnel test conducted to determine the viscosity of the SCC varied between 7 and 16 s. While the mixtures containing 30% GBFS and MD in place of fine aggregate were in viscosity class VF2, the other mixtures were found to be in the VF1 viscosity class. The results of the V-funnel test suggested that the GBFS20 mixture is 15% more viscous than the Reference specimen. The L-box test, measuring the filling ability of the SCC, produced passing ability values in the range of 0.80–0.91 and were thus found to be suitable for the PA2 class. These results indicate that the risk of blocking by grains in the MD30 mixture was higher than with the other mixtures. It is believed that the specific surface area of MD being higher than that of GBFS contributed to the increased viscosity and decreased consistency.

#### 4.2.2. Rheological Analyses

The rheological properties of the SCC mixtures were determined using a BT-2 concrete rheometer, the results of which are given in Table 5 and Figure 5. The lowest viscosity among the SCC mixtures produced using MD was obtained from the MD10 mixture.

According to the test results, the GBFS30 mixture had the highest viscosity value among the SSC mixtures containing GBFS, which was attributed to the grain structure of GBFS. The results suggest that the use of GBFS in ratios of up to 20% contributed to the improvement of both viscosity and yield stress values when compared to the reference specimen. These results are consistent with those of the workability tests, and it is thus thought that the use of GBFS in SCC mixtures can bring various advantages in terms of, for example, aggregate arching and internal friction.

### 4.3. Hardened Concrete Tests

#### 4.3.1. Compressive Strength Test Results

The mechanical properties of the SCC mixtures were determined through uniaxial compressive strength tests performed on hardened concrete specimens, the results of which are presented in Figure 6. Three cube specimens were tested for each age and average values were given in Figure 6. Considering the early-age compressive strength of the specimens, the MD did not contribute to strength when compared to the reference specimen, while the specimens containing GBFS showed improvement at an early age when compared to the reference specimen.

Similar behavior was observed in the 28-day strength test results, with only the MD10 series specimen exhibiting greater strength than the reference specimen among those containing MD. As to late-age strengths, the highest compressive strength was achieved with GBFS30 due to the pozzolanic feature of the GBFS additive. To obtain the best possible strength, the optimum ratio of MD is 10%, while GBFS can be used in ratios of up to 30%.

#### 4.3.2. Abrasion Resistance Test Results

The results of the abrasion resistance test are presented in Figure 7, in which it can be seen that the SCC mixtures with additives performed better than the reference specimen. The best abrasion performance of the specimens with MD was achieved by the MD10 mixture at 10.34%. Among the mixtures containing MD, the highest abrasion value was observed in the MD30 series, being 13.47% higher than the Reference mixture. Fine MD is thought to lead to a loss of workability, thereby increasing the porosity of hardened SCC, and the results of the compressive strength test support this finding. The best abrasion resistance performance among the additives used was noted in the specimens containing GBFS, with GBFS30 recording the highest abrasion resistance and 26.78% less abrasion than the reference specimen. It was thus concluded that the GBFS series performed better than the MD series, especially in its forming of additional CSH gels as a result of a reaction with the Ca(OH)_2_ that formed as a result of cement hydration. This was attributed to its pozzolanic nature as well as its fine-grained structure and gap-filling ability.

#### 4.3.3. Freeze–Thaw Resistance Test Results

At the end of the freeze–thaw test, no visible physical deterioration or weight loss was noted in the concrete specimens, which was attributed to the high amount of cement and the mineral additives used. The results of the compressive strength tests performed on the concrete specimens after the freeze–thaw test are presented in Figure 8. They indicate that the strength losses at the end of the freeze–thaw cycles varied between 2% and 10%.

What is worthy of note here is that the strength losses increase as the ratio of additive increases in the mixtures containing MD. Previous studies have shown that a replacement ratio of more than 15% when using such stone dust as marble increases the freeze–thaw resistance losses [52]. This may be due to the fact that the MD additive used in the SCC mixture alters the pore pattern in the composite structure. The strength losses in the mixtures containing GBFS varied between 2% and 6%, and it is thought that the gap-filling ability and the pozzolanic reaction of GBFS contributed to the lowest loss of strength, namely, 2% in the GBFS30 specimen. After the freeze–thaw cycles, a pore analysis was conducted on the concrete specimens through mercury porosimetry, the results of which are presented in Table 6. The results show that the lowest porosity and total pore area after the freeze–thaw test were obtained from the MD10 series among the SCC specimens with MD and that the pore area and porosity exceeded those of the reference specimen with the increased use of MD.

It can be concluded that the strength values decreased due to the weakening of the bond between the cement paste and aggregate interface within the composite structure as the porosity increased. In the mixtures containing GBFS, the lowest porosity and pore ratio values were obtained after the freeze–thaw test, among which the porosity of the GBFS30 mixture was 57% lower than the reference specimen. This also explains the strength loss behavior observed in the results of the compressive strength test.

#### 4.3.4. Sulfate Resistance Test Results

Mortar sticks prepared to determine the sulfate resistance of the SCC specimens were produced without the use of coarse aggregate, and with MD and GBFS used in place of fine aggregates in different proportions. The changes in the length of the prism specimens conditioned in a solution of 10% Na_2_(SO)_4_ for a total of 6 months are presented in Figure 8. The maximum change in length, measured with a precision comparator, was 0.0185% in the reference specimen containing no mineral additives, although this value was below the limit value of 0.1% defined in ASTM C 1012 [49]. The high cement dosage, the proportion of mineral additives and the pore-free structure due to high workability contributed to the low sulfate expansion in all SCC specimens.

The GBFS30 series achieved the best performance in the sulfate test (0.0041%). The sulfate resistance of GBFS, which is used in the production of sulfate-resistant cement in the cement industry, played a key role in achieving this result. Furthermore, thanks to the improved workability associated with the GBFS additive, the obtained specimens were less porous, making it difficult for harmful chemicals to infiltrate the composite structure, thereby reducing potential damage. The changes in the length of the specimens containing MD after the sulfate test were also below the limit value, although it was observed that the rate of change in length was greater than in the specimens containing GBFS (Figure 9). It is thought that this was due to the lack of a pozzolanic feature in MD and that its chemical structure improved its sulfate resistance. The lowest rate of change in length among the MD series samples was obtained in specimen MD10.

### 4.4. Microstructural Investigation

The internal structure and pore diameters of the reference specimen obtained from the SEM analyses of the sections extracted from the produced SCC specimens are presented in Figure 10. Looking at the internal structure of the reference specimen, it was apparent that a porous structure was dominant across the composite. The largest pore diameter observed in specimen MD10 was as 172.6 µm—a value approximately one-third that of the reference specimen. However, the largest pore diameter among the SCC series was found in the MD30 specimen, with 553.3 µm.

The lowest pore diameter among the SCC mixtures was observed in mixture GBFS30, which, as can be seen in Figure 9, was 86.67 µm. A comparison of this result with the results of the compressive strength test indicates that the high compressive strength of the GBFS30 series was attributable to the low pore diameter and low pore ratio. Similarly, the small pore distribution of the GBFS30 series, as seen in the results of the abrasion test in Figure 7, resulted in the best abrasion resistance in this specimen. The largest pore diameter among the GBFS series, at 386.9 µm, was obtained in the GBFS10 series, as seen in Figure 10, and its porosity was found to be 10.09% in the mercury porosimetry test results (Table 6). A general evaluation of the internal structure images indicates that the SCC containing GBFS had smaller air pore diameters than the reference specimen, thanks to its high workability. As to the effect of the type of additive on pore diameter, it was observed that the GBFS additive filled better than MD, and it was concluded that the pozzolanic feature of GBFS contributed to this result.

The internal structure images of the SCC specimens following the sulfate resistance test are shown in Figure 11. The images clearly show the formation of acicular ettringite crystals in the internal structure of the reference specimen, which contained no mineral additives. Ettringite crystals, which expand in the composite in the presence of humidity, can cause damage to the concrete over time. Taking into account the length change results from the sulfate test (Figure 11), this ettringite formation in the reference specimen explains the largest rate of change in length in the reference specimen. In the SCC series, the largest ettringite formation was observed in the MD30 specimen. The large formation of ettringite in this image and the rates of change in length in Figure 9 explain why the greatest change was recorded in length after the reference specimen was observed in specimen MD30 under sulfate attack. GBFS is a mineral additive that is known to perform well against sulfate attack. A general examination of the internal structures showed ettringite formations to be almost non-existent in specimens containing GBFS. Following the sulfate test, an examination of the internal structures of the specimens containing GBFS revealed very small ettringite crystals and portlandite plaques in specimen GBFS10. A 3000× magnified image of specimen GBFS20 revealed ettringite formations in its internal structure. As is the case with specimen GBFS20, specimen GBFS30, containing the highest ratio of GBFS, also had a low rate of ettringite crystal formation. These examinations, together with the rates of change in length given in Figure 9, explain this behavior in GBFS20 and GBFS30, which recorded the lowest rates of expansion in the sulfate test.

## 5. Discussion

GBFS is a type of pozzolanic industrial waste while MD has no pozzolanic behavior. The reduction in strength for MD used as supplementary cementitious materials is related to the dilution of pozzolanic reactions in these mixtures [53,54]. According to this phenomenon, this research is focused on the utilization of MD as an aggregate replacement against the fine aggregate portion of the SCC. In the study, it was also aimed to determine the influence of filling and pozzolanic abilities of the GBFS usage on the desired workability, durability and mechanical properties of SCC as prescribed in the previous studies [55]. According to fresh SCC results, the optimum MD usage is found as 10% while this replacement ratio is 20% for the GBFS. This fact depends on the fineness of the MD and GBFS as presented in Table 3. This is because the higher specific surface area of the MD shows a better-filling effect in the matrix; however, replacement over 10% causes an increase in the viscosity and reduction in the workability. Higher MD and GBFS replacement against fine aggregate had increased the total powder content of the mixture which caused an increase in plastic viscosity [17]. The filling ability (slump flow and T_500_) and passing ability (L-box testing) have been determined by standard workability tests [42]. The results revealed that aggregate replacement ratios above 20% reduced the filling and passing abilities of the SCC mixtures. This result can be attributed to the irregular-shaped particles from GBFS and MD, leading to the reduction in the friction factor and enhancement in the workability performance [51]. In addition, increasing the fineness of the total particles in the SCC mixture can increase the amount of mixing water for the required consistency, which could lead to an increase in the porosity of the concrete [56].

Compressive strength test results showed different behavior in both MD and GBFS specimens due to the increased aggregate replacement ratios. Belaidi et al. had found that 10% MD usage in binary SCC mixtures showed optimum performance for both workability and compressive strength [57]. This behavior can be explained by the non-pozzolanic structure of the MD. The strength increase for the MD10 mixture depends on the filler effect of the fine MD that is replaced against fine aggregate in the SCC mixture [58,59]. The dual effect of GBFS (filler and pozzolanic) that is replaced as aggregate resulted in a more effective strength performance than MD thanks to both the filling and the pozzolanic effect playing a role in the compressive strength of SCC specimens. The strength development of SCC specimens using GBFS are positively affected at later ages due to the reactive silica content as defined in the previous studies [60,61].

The same effect of MD and GBFS also influenced the durability properties of the hardened composite. Unit weight and abrasion resistance of the produced SCC specimens have been beneficially affected by the gap-filling effect of the MD and GBFS [26]. Additionally, the pozzolanic reaction products inside the matrix increase the durability by reducing the amount of cement hydration by-product Ca(OH)_2_ especially for sulfate resistance of SCC [62].

## 6. Conclusions

The novelty of this study is focused on the replacement of MD and GBFS against the fine aggregate part of the SCC mixture. The results of the workability test indicate that the use of MD at a ratio above 10% has an adverse effect on workability. The loss of workability in the specimens containing GBFS was 30% and above. It was observed that GBFS contributed more to compressive strength than MD. The highest strength performance was achieved by specimen GBFS30, which was attributed to its pozzolanic effect. MD, on the other hand, recorded strengths above that of the reference specimen when used in a ratio of 10% thanks to its gap-filling ability and the stronger bond between the aggregate and the cement paste (ITZ). The results of the durability tests suggested that GBFS has the potential to contribute significantly to the production of durable SCC. MD, on the other hand, positively affects the durability of SCC when used in ratios of up to 20% in place of fine aggregate. An analysis of the microstructure and porosity of the SCC specimens indicated that the addition of MD and GBFS contributed significantly to the filling of pores. It was also observed that the use of GBFS against sulfate attack had a reducing effect on the formation of ettringite crystals in the internal structure of the SCC. The results indicate that the use of MD and GBFS as filler materials in the production of SCC positively affects the mechanical and durability properties of concrete. The use of MD and GBFS in SCC production will both increase concrete performance and offer new opportunities for the utilization of waste that would otherwise cause environmental problems. In addition, the usage of these industrial wastes against fine aggregate will help for the consumption of high amounts of MD and GBFS in SCC mixtures when compared with usage as a supplementary cementitious material. However, the transportation and screening of these wastes to the construction site can increase the cost of the SCC. Future studies can focus on the compatibility of mineral additives and powder materials that can increase viscosity, as materials that are being used increasingly in SCC technologies with plasticizer additives in support of strength and different durability properties apart from this study. Moreover, binary and ternary replacement of MD and GBFS with other wastes such as fly ash and construction demolish wastes can be investigated for obtaining more effective solutions against the consumption of these wastes in concrete technology.

## Figures and Tables

**Figure 1 materials-15-01795-f001:**
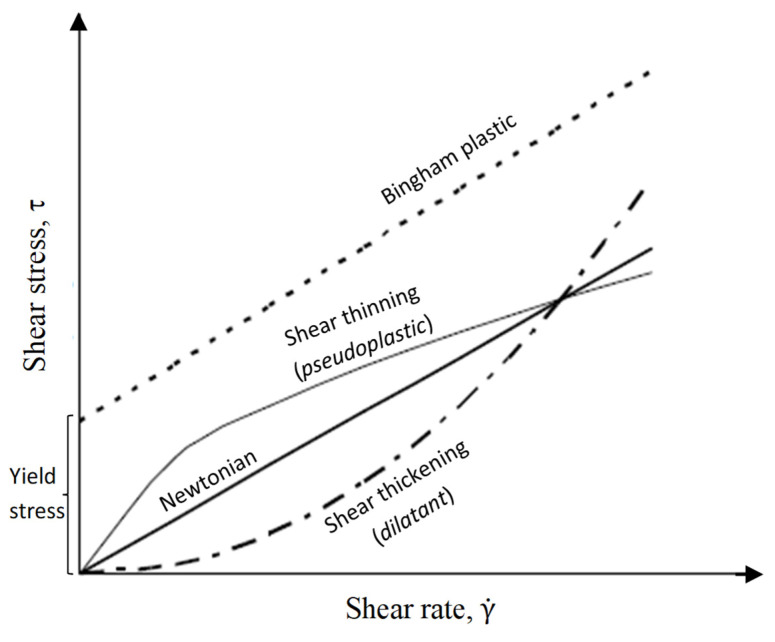
Shear stress-dependent on the shear rate in fluids.

**Figure 2 materials-15-01795-f002:**
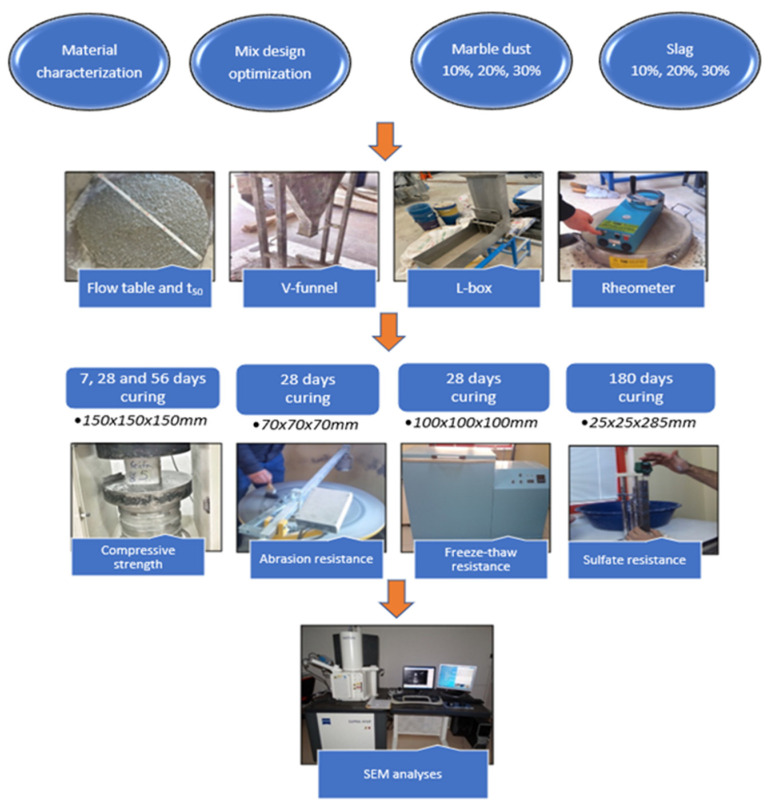
Schematic presentation of the study process.

**Figure 3 materials-15-01795-f003:**
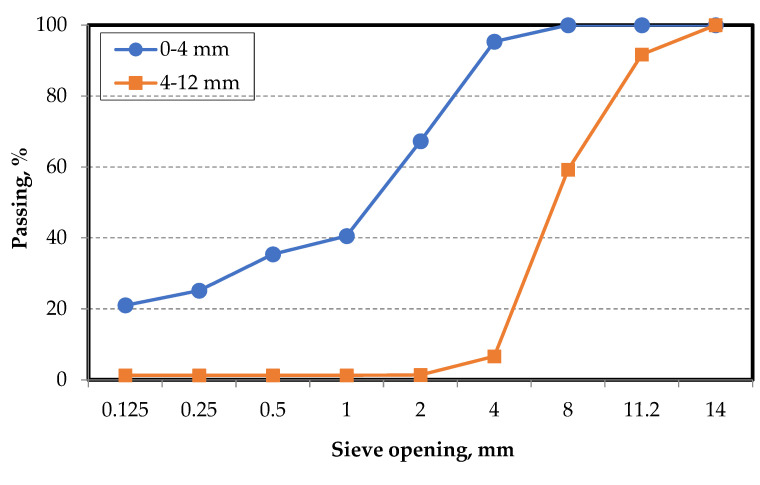
Particle size distribution of the aggregates.

**Figure 4 materials-15-01795-f004:**
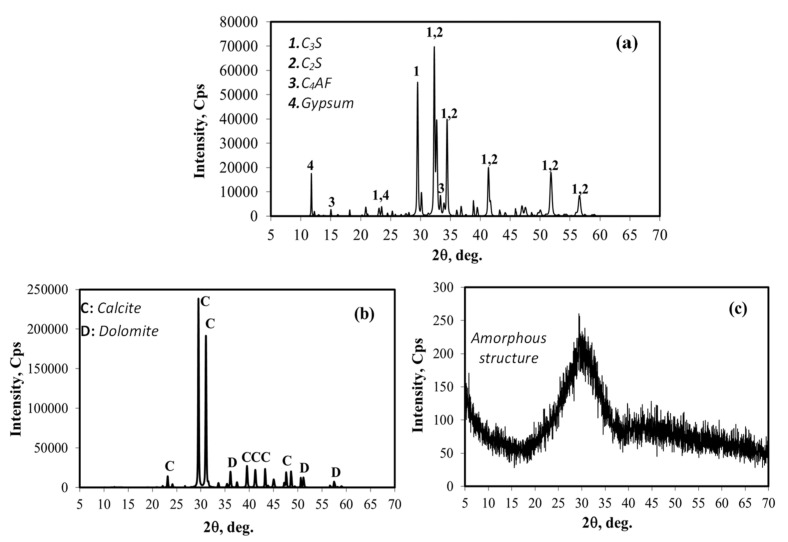
XRD analyses of the materials: (**a**) CEM I 42.5 R, (**b**) MD, (**c**) GBFS.

**Figure 5 materials-15-01795-f005:**
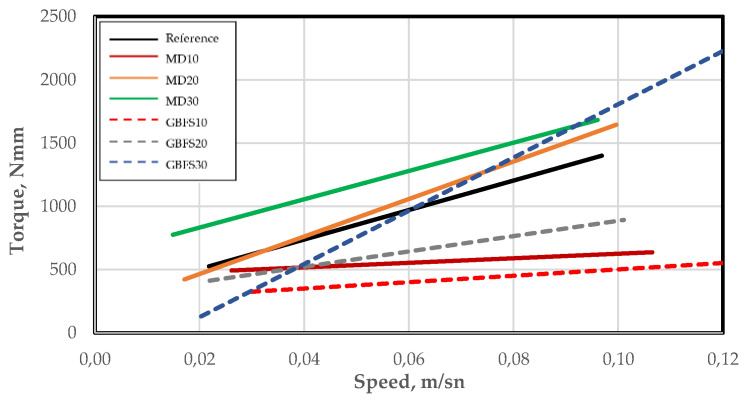
Rheometer analyses of the SCC mixtures.

**Figure 6 materials-15-01795-f006:**
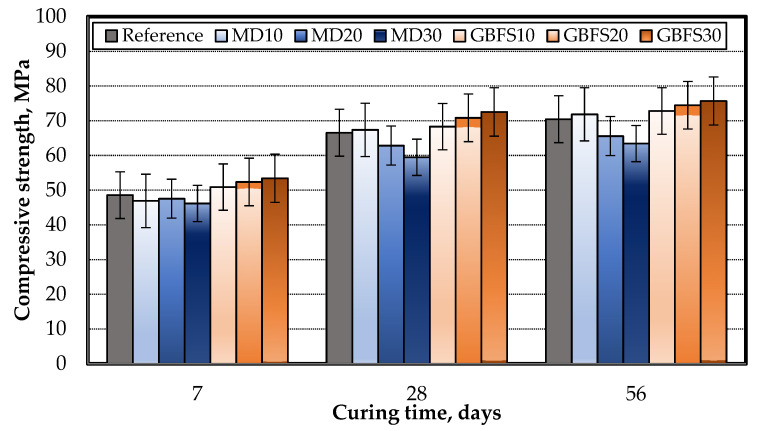
Compressive strength of the SCC specimens.

**Figure 7 materials-15-01795-f007:**
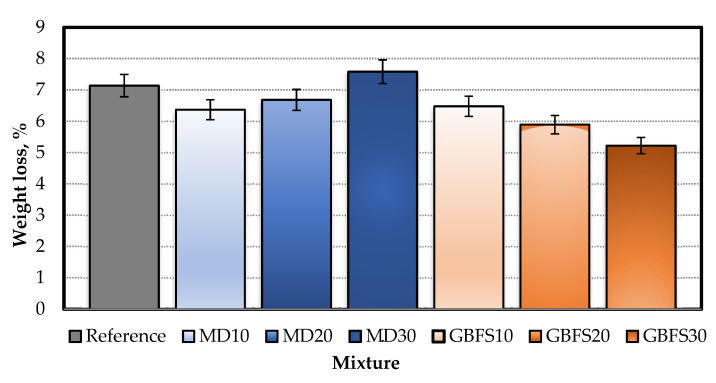
Bohme abrasion test results.

**Figure 8 materials-15-01795-f008:**
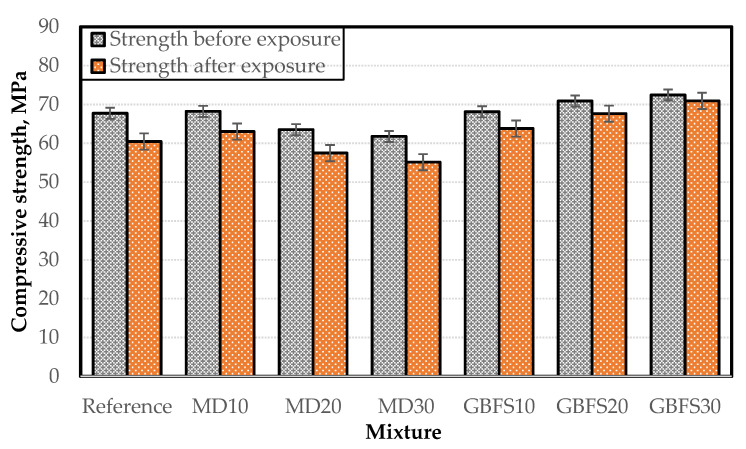
Strength behavior of the mixtures after freeze–thaw cycles.

**Figure 9 materials-15-01795-f009:**
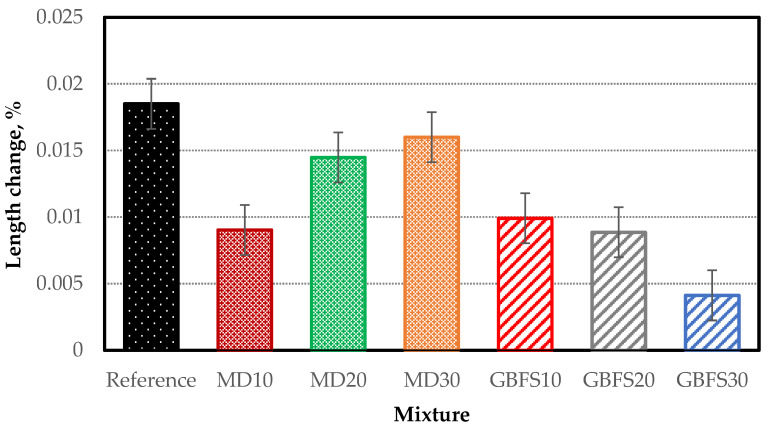
Length chance of the specimens after sulfate exposure.

**Figure 10 materials-15-01795-f010:**
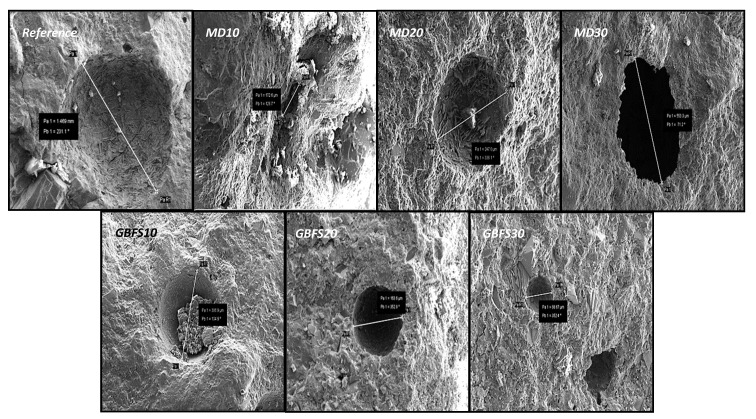
The pore structure of the SCC mixtures.

**Figure 11 materials-15-01795-f011:**
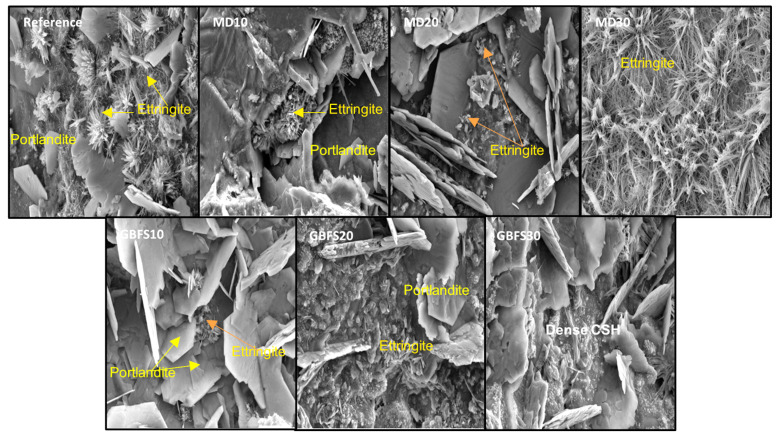
Microstructure of the SCC specimens after sulfate effect.

**Table 1 materials-15-01795-t001:** Mix proportion of the SCC mixtures for 1 m^3^.

Mixture	Cement	Water	0–4 mm	4–12 mm	MD	GBFS	Plasticizer
	kg	kg	kg	kg	kg	kg	kg
Reference	380	200	950	850	-	-	7
MD10	380	200	888	850	62	-	7
MD20	380	200	826	850	124	-	7
MD30	380	200	764	850	186	-	7
GBFS10	380	200	888	850	-	62	7
GBFS20	380	200	826	850	-	124	7
GBFS30	380	200	764	850	-	186	7

**Table 2 materials-15-01795-t002:** Chemical analyses of the raw materials.

Chemical Composition, %			
Property	Cement	MD	GBFS
CaO	64.53	53.35	32.43
SiO_2_	21.22	0.61	39.64
Al_2_O_3_	4.67	0.23	12.24
Fe_2_O_3_	2.91	0.13	0.72
MgO	0.96	0.79	7.51
SO_3_	2.92	0.54	1.43
MnO	-	-	1.44
K_2_O	0.72	0.03	1.58
Na_2_O	0.20	0.69	0.38
TiO_2_	-	-	0.61
LOI	2.8	43.07	-
C_2_S	11.13	-	-
C_3_S	57.58	-	-
C_3_A	7.45	-	-
C_4_AF	8.86	-	-

**Table 3 materials-15-01795-t003:** Physical properties of the raw materials.

Material	Specific Surface Area	Density
	cm^2^/g	g/cm^3^
Cement	3382	3.062
MD	3930	2.750
GBFS	3527	2.919

**Table 4 materials-15-01795-t004:** Workability test results of the SCC mixtures.

Mixture	Slump Flow	T_500_	V-Funnel	L-Box
	mm	s	s	PA
Reference	650	4	11	0.87
MD10	710	4	8	0.88
MD20	670	5	9	0.84
MD30	550	6	18	0.80
GBFS10	680	4	10	0.88
GBFS20	720	3	7	0.91
GBFS30	670	5	13	0.82

**Table 5 materials-15-01795-t005:** Rheological parameters of the SCC mixtures.

Mixture	Plastic Viscosity	Yield Stress
	Nmms	Nmm
Reference	11,650	272
MD10	1795	168
MD20	11,179	446
MD30	14,811	608
GBFS10	2522	250
GBFS20	6060	280
GBFS30	21,031	297

**Table 6 materials-15-01795-t006:** Mercury intrusion porosimeter results of the SCC specimens.

Mixture	Total Pore Area	Average Pore Diameter	Porosity
	m^2^/g	µm	%
Reference	4.048	0.0604	10.49
MD10	3.765	0.0442	9.74
MD20	4.986	0.0471	11.99
MD30	5.271	0.0412	12.96
GBFS10	3.805	0.0744	10.09
GBFS20	2.575	0.0390	8.49
GBFS30	1.314	0.0820	4.44

## Data Availability

Not applicable.

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
