# Peer review of "Rheological and Durability Properties of Self-Compacting Concrete Produced Using Marble Dust and Blast Furnace Slag"

_materials, 2022, doi:10.3390/ma15051795_

Round 1

Reviewer 1 Report

It was a great effort by the author's on implementing this research work. It shows that authors focusing on the studies on SCC with influence of marble dust (MD) and blast furnace slag (GBFS) as replacement to aggregate 0-4mm.

Here few recommendations for authors to improve their manuscript further.

  1. There are many researches published influence of MD and GBFS in SCC. So, what makes difference and novelty of this work from previous studies. Also, authors have to focus brief discussion in the Introduction part.
  2. It is better to make separate discussion of research gap and novelty of their work.
  3. Authors have to give proper reason and justification of slump flow results. Because when compared to reference SCC mix with MD10 shown improvement of flow and other mix combinations (MD20, MD30) shown reduction of flow. Why? Whereas in GBFS10 and GBFS20 shown improvement in flow. And why GBFS30 has shown reduction of flow?
  4. Other workability parameters also shown related results. Why?
  5. Compressive strength results also differs. Why? Justify them.
  6. Just discussing achieved results of all properties in Results and Conclusions part is not enough. Authors have to give proper reasons to achieve those results and required to justify them.

After rectifying the major revisions of the manuscript, it can be reconsidered.

Author Response

The authors would like to thank you for your kind and constructive comments. The detailed responses to your comments and revised manuscript files have been uploaded. Please see the attachment.

Reviewer 2 Report

The manuscript presented a well-organized research paper. The paper is very interesting and worth publication in the Materials in the reviewer's view. However, following comments are advised to be considered before acceptance:

Introduction:
Major Comment: The introduction is lack of sufficient background information which is unable to give the reader detailed background knowledge and possible wide application of this study. The literature review is very poor. The introduction needs to be more emphasized on the research work with a detailed explanation of the whole process considering past, present and future scope. How the present study gives more accurate results than previous studies? It needs to be strengthened in terms of recent research in this area with possible research gaps. It is strongly recommended to add a recent literature. Research gaps should be highlighted more clearly and future applications of this study should be added.

Try to refer to some recent and up-to-date research papers, especially 2020 and 2021.

Your introduction is too short. Please refer to some new papers in this area.

Try to refer to some recent and up-to-date research papers related to the topic, especially in recent years like those related to research related to waste materials like:

  • 3390/ma15020430

Experimental program

Please report enough detail in order to replicate the study.

Plots of all Figures need to be uniformed in size and style
Did you repeat the experimental tests to assure the results?
Please mention every standard you used in this study.
Explain the test set up in more detail.
Please explain the sample preparation in more detail.

Marble and GBFS are used as fine aggregate. I think both are waste materials and function are same. It would be better if one of them was used as binding materials.

Specific gravity of marble and GBFS are different from fine aggregate. Then how you can replace it by weight. Please explain this in detail.

Conclusions

Please mention your study limits and suggest some future research topics.
The authors are advised to write the conclusion in a comprehensive way, it should contain key values, suitability of the applied method, contributions and possible future work.

The "Conclusions" may be provided as a "uniform text", rather than using bullet points. The weaknesses of the work and the future improvements should be added in this section

Author Response

(The authors gave the same response as above.)

Reviewer 3 Report

Pls. find the attached comments

Author Response

(The authors gave the same response as above.)

Reviewer 4 Report

Ms. Ref. No.: materials-1591282

Rheological and Durability Properties of Self-Compacting Concrete Produced Using Marble Dust and Blast Furnace Slag

Reviewer comments:

SUMMARY

The manuscript deals with a good study on the properties of self-compacting concrete produced using marble dust and blast furnace slag. Furthermore, this is a topic that has not been widely covered in the literature, therefore, this a subject of great interest, but it is somehow limited in the analysis and application of these results.

MAIN IMPRESSION

This paper has an undeniable practical usefulness. However, from a scientific point of view, the following issues must be addressed: i) This topic is in itself a matter that requires further discussion, ii) It is necessary to interpret and describe the significance of your findings in light of what was already known and iii) It is recommended to underscore the novelty of this paper in the conclusion.

MORE DETAILED COMMENTS

Line 1: It should be “Article” instead of “Type of the Paper (Article)”.

Line 22: Could you please add ”marble dust (MD)”.

Line 29: Fluidity is the quality or state of being fluid or the physical property of a substance that enables it to flow. Then, what do you mean with “rate of fluidity”?

Lines 33&34, 45&46 and so on: Distance between paragraphs is wrong. You should follow the format rules.

Line 35: Probably, a dot is missing after ”concrete [3] “. Could you please check the format throughout the text?

Line 99, 104, 107 and 108: Could you please add “Turkey”?

Line 98: Could you please add the cement, GGBFS, marble 2.1. Materials

Line 131, 132 and so on: C° is wrong.

Line 115: Could you please add a reference for “EFNARC”?

Line 153: Could you please add the Na2O content in ”Table 2. Chemical analyses”?

Line 159: Could you please provide the C3S, C2S, C3A, C4AF content?

Line 165: 39.64% is the SiO2 total content. Could you please add the reactive SiO2 content?

Line 342: paper discussion should be added in 3. Results (3. Results and discussion) or in a new chapter (4. Discussion). “The purpose of the discussion is to interpret and describe the significance of your findings in light of what was already known about the research problem being investigated, and to explain any new understanding or fresh insights about the problem after you've taken the findings into consideration.

Lines 138-341: Only three references are used to disscuss the results. ”The discussion will always connect to the introduction by way of the research questions or hypotheses you posed and the literature you reviewed, but it does not simply repeat or rearrange the introduction; the discussion should always explain how your study has moved the reader's understanding of the research problem forward from where you left them at the end of the introduction.”

Lines 382 – 466: References

References must be listed individually at the end of the manuscript as follows:

  1. Author 1, A.B.; Author 2, C.D. Title of the article. Abbreviated Journal Name Year, Volume, page range.

For instance, ref. [1]  1. Wu, M.; Xiong, X.; Shen, W.; Huo, X.; Xu, G.; Zhang, B.; Li, J.; Zhang, W. Material Design and Engineering 383 Application of Fair-Faced Self-Compacting Concrete. Construction and Building Materials 2021, 300, 123992, 384 doi:10.1016/j.conbuildmat.2021.123992, is wrong.

It should be:  … Constr. Build. Mater.

RECOMMENDATION

In conclusion, Major changes have been proposed.

Author Response

(The authors gave the same response as above.)

Round 2

Reviewer 1 Report

I appreciate the authors for modifying the manuscript as per the comments accordingly.

Author Response

As the authors, we thank you very much for your comments that contributed to the improvement of the quality of the study.

Regards 

Reviewer 4 Report

Ms. Ref. No.: materials-1591282-peer-review-v2

Rheological and Durability Properties of Self-Compacting Concrete Produced Using Marble Dust and Blast Furnace Slag

Reviewer comments:

The following comments have not been considered:

  • Line 131, 132 and so on: C° is wrong. This was not corrected: line 167: -20 C° and 20 C° [48],
  • Line 153: Could you please add the Na2O content in ”Table 2. Chemical analyses”?
  • Line 159: Could you please provide the C3S, C2S, C3A, C4AF content?
  • Line 165: 64% is the SiO2 total content. Could you please add the reactive SiO2 content?

Could you please explain why not?

References

References must be listed individually at the end of the manuscript as follows:

  1. Author 1, A.B.; Author 2, C.D. Title of the article. Abbreviated Journal Name Year, Volume, page range.

For instance, ref. [3]  3. Brouwers, H.J.H.; Radix, H.J. Self-compacting concrete: theoretical and experimental study. Cement Concrete Res. 2005, 35, 467 2116–2136. , is wrong.

It should be:  … Cem. Concr.Res.

MAIN IMPRESSION

  1. i) It is necessary to interpret and describe the significance of your findings in light of what was already known and ii) It is recommended to underscore the novelty of this paper in the conclusion.

RECOMMENDATION

In conclusion, Major changes have been proposed.

Author Response

The authors would like to thank you for the constructive and guiding suggestions of the reviewer comments. The responses to reviewer comments and revised version of the manuscript have been attached to the manuscript tracking system. 
